# Attrition and associated factors among patients on chronic antihypertensive therapy at Mulago hospital, Uganda: A mixed method study

Nathan Ntenkaire[1,2]*, Mark Kaddu Mukasa[3], Patience Muwanguzi[1,4], Brian Mikka[1], Sandra Lunkuse[1,5], Julius Mubiru[1], Maxwell Okwero[1,2], Beatrice Basuuta[1], Douglas Bulafu[1,6], Joan N. Kalyango[1,7]

1 Clinical Epidemiology Unit, College of Health Sciences, Makerere University, Kampala, Uganda, 2 Infectious Diseases Institute, College of Health Sciences, Makerere University, Kampala, Uganda, 3 Department of Internal Medicine, School of Medicine, College of Health Sciences, Makerere University, Kampala, Uganda, 4 Department of Nursing, School of Health Sciences, College of Health Sciences, Makerere University, Kampala, Uganda, 5 The Medical Research Council/Uganda Virus Research Institute, and LSHTM research unit, Entebbe, Uganda, 6 Department of Disease Control and Environmental Health, School of Public Health, College of Health Sciences, Makerere University, Kampala, Uganda, 7 Department of Pharmacy, School of Health Sciences, College of Health Sciences, Makerere University, Kampala, Uganda

* michealnathans@gmail.com

## Abstract

### Background

Attrition among patients on chronic antihypertensive therapy is a significant problem that can lead to serious health consequences, including uncontrolled blood pressure. Several factors underlie attrition, so healthcare providers must address them to prevent treatment discontinuation and ensure optimal outcomes. Therefore, this study assessed attrition and associated factors among hypertensive patients from January 2020 and December 2022.

### Methods

A sequential explanatory mixed-methods design. The quantitative study was a retrospective cohort study design using files of 1215 hypertensive patients. The qualitative study employed an explanatory descriptive design among 16 patients. A data abstraction tool and an interview guide were used for data collection. Attrition was defined as patients who were lost to follow-up. Extended Cox regression was used to determine the factors associated with time to attrition at 5% level of significance and qualitative data analysis employed a thematic analysis codebook.

### Results

The attrition proportion was 56.8% (95% confidence interval (CI) 54.0–59.7) with most patients getting lost to follow-up in 2020 (64.9%) and fewest in 2021 (54.7%).

**Data availability statement:** All data and materials underlying the findings of this study are available in a public repository. The anonymized dataset, stata dofile, and data dictionary have been deposited in Zenodo and can be accessed at https://doi.org/10.5281/zenodo.17553717.

**Funding:** The author(s) received no specific funding for this work.

**Competing interests:** The authors have declared that no competing interests exist.

Age (hazard ratio (HR)=0.947, 95% CI 0.931–0.963), female Sex (HR = 0.734, 95% CI 0.620–0.869), residence outside Kampala (Capital City) (HR = 1.24, 95%CI 1.063–1.455), 2022 cohort entry year (HR = 1.433, 95% CI 1.156–1.777), last visit systolic blood pressure (SBP) (HR = 1.014, 95% CI 1.009–1.018), and last visit diastolic blood pressure (DBP) (HR = 0.947,95% CI 0.931–0.963) were associated with time to attrition. Loss to follow-up (LTFU) was driven by structural and contextual barriers, health system challenges, and illness perceptions and health-related limitations.

## Conclusion

Hypertensive patient attrition proportion is high, below the Centers for Disease Control and Prevention's 80% retention target. This calls for innovative retention strategies, and targeted support for high-risk groups like the male patients and those distant from the health facility. Patient-centered approaches addressing structural and health system barriers are essential to improving retention in hypertension care.

## Introduction

Non-communicable diseases (NCDs) are responsible for the majority of deaths worldwide, low and middle-income countries (LMICs) contribute to approximately 75% of all NCD-related fatalities [1]. Globally, an estimated 1.28 billion adults (30–79 years old) have hypertension, and the majority (two-thirds) of these live in LMICs [2]. In sub-Saharan Africa (SSA), the prevalence of hypertension has risen, with estimates in 2019 reaching 48% (CI: 42–54%) among women and 34% (CI: 29–39%) among men [3]. In Uganda, hypertension is estimated to have a prevalence rate of 26.4% in general, with the highest rate of 28.5% occurring in the central region, and the lowest rate of 23.3% in the northern region. [4]. Furthermore, the prevalence rates of hypertension in urban and rural areas are estimated to be 28.9% and 25.8%, respectively [5]. Only 3.6% of hypertensive patients in Uganda have their blood pressure (BP) under control, and hypertension (HTN) and other NCDs account for 33% of all deaths with a 22% probability of dying prematurely from either cardiovascular disease, cancer, chronic respiratory diseases or diabetes [6]

Although HTN cannot be cured, it can be controlled with medication, dietary changes, or a combination of both [7]. Patients who consistently engage in medical care at a healthcare facility are considered to be retained in hypertension care. For the long-term management of hypertension and program maintenance, it is critical to have high retention rates. However, in resource-limited settings, 1-year retention rates are often below 50% [8]. Retention on antihypertensive therapy is essential for controlling hypertension and lowering the risk of complications related to high BP. Hypertension can lead to various complications, such as cardiovascular disease, kidney disease, cognitive impairment, and eye damage. These complications can cause a range of burdens, including higher healthcare expenses, reduced productivity and quality of life, and an increased risk of disability and premature death [9].

Attrition from antihypertensive therapy varies in different studies. Studies done in Cambodia and India reported attrition ranging from 9.2% to 61.5% [10,11]. The factors associated with attrition include patient-related factors such as sex, smoking, age, body mass index (BMI). illiteracy, low income, multi-person household [12,13]. There are also clinical factors such as uncontrolled BP, adverse events, medication regimen complex index, and history of hospitalization [14,15]. In addition, health system factors such as stock out, quality of health services, physician-patient relationship, health education and availability of medication have been associated with attrition [16]. With the rising prevalence of chronic diseases, health systems in Africa are struggling to maintain continuity of care [17], yet no research has been conducted in Uganda to determine the attrition proportion of patients with hypertension and it is hypothesized that attrition proportions are high and its predictors are varied. Additionally, patients view on the reason for LTFU from hypertension care is less known, Therefore, this study determined attrition and associated factors among hypertensive patients and explored reasons for LTFU from the perspective of patients identified as lost to follow-up.

## Materials and methods

### Study setting

The study was conducted at the hypertension clinic in the medical outpatient's department of Mulago Hospital, which is situated in Kampala, Uganda's capital city. The hospital serves as the primary teaching facility for the College of Health Sciences at Makerere University. It offers comprehensive healthcare services across numerous medical and surgical sub-specialties, including dentistry, emergency medicine, pediatrics, and intensive care. The clinic is open on Mondays, except on public holidays, and is staffed by specialist physicians, including a cardiologist, nurses, and records officers as well as laboratory and pharmacy support. Approximately 438 patients are initiated on antihypertensive treatment (AHT) each year at this clinic.

### Study design and population

A sequential explanatory mixed methods design, consisting of two distinct phases by Tashakkori and Creswell [18] was adopted. The quantitative study phase was a retrospective cohort, and the qualitative study phase was a descriptive explanatory design. For the quantitative study, files of hypertensive patients who were registered at the Mulago hypertension clinic between January 2020 and December 2022 were included. Patients' files missing data on more than 30% of the studied variables were excluded. Patients who were registered between January 2020 and December 2022, who were confirmed as lost to follow up and consented to participate were included in the qualitative study. Patients who were transferred to other health facilities and those who did not understand English or Luganda were excluded from participation in the interviews.

### Sample size and sampling procedure

The sample size was determined to address two objectives: (i) to determine the attrition proportion and (ii) to determine factors associated with time to attrition among hypertensive patients. For determining the attrition proportion, the Kish (1965) formula for determining sample size was initially applied [19]. Using an attrition proportion of 9.2% among hypertensive patients reported by Meena and colleagues in India, at a 95% confidence level and a 5% margin of error [20]. A minimum sample size of 397 of patients file was required

To determine the factors associated with time to attrition, the sample size was estimated using the formula for survival data (S1 Appendix). Assuming a two-sided 5% level of significance, 80% power, and proportions of patients retained in care of 67.9% of the 53 patients aged 41–52 years and 79.6% of the 54 patients aged 53–65 years as reported by Given et al. (1985) [21].

Therefore, the minimum number of patient files required was 1006 after accounting for 10% missing data. Much as the overall calculated minimum required sample size was 1,006, the study employed a consecutive sampling approach, in

which all 1,215 patient files that met the eligibility criteria during the study period (January 2020–December 2022) were included in the analysis.

Purposive sampling with maximum variation was used to select participants for the qualitative study, considering factors such as age, sex, residence, presence of comorbidities, and adverse events, from the 690 patients identified as lost to follow-up. Interviewing of participants was stopped at the point where additional interviews no longer yielded new insights and adequate depth of understanding of the topic had been obtained.

## Variables

The dependent variable was time to attrition. Attrition was defined as patients who were lost to follow-up and loss to follow up was referred to patients who missed ≥2 consecutive clinic appointments. Independent variables consisted of patient-related factors, clinical factors, and health system factors. Patient-related factors included age, sex, residence, occupation, alcohol use, smoking status, herbal medicine use, and marital status. Clinical factors comprised baseline systolic and diastolic blood pressure, presence of comorbidities, history of hospitalization, number of antihypertensive medications, reported side effects, cohort entry year, the antihypertensive regimen, and last-visit systolic and diastolic blood pressure. The health system factor assessed was the occurrence of medicine stock-outs.

Follow-up appointments varied by patient, with each given an individualized schedule based on clinical condition and treatment response. For survival analysis, the time origin was the date of registration at the clinic, and person-time was calculated from this date until attrition and those without attrition were censored at the date on which data was collected.

## Data collection

Secondary data obtained in the hospital for patient's clinical monitoring and evaluation was used in this study and it was accessed between 07th July 2023–20th November 2023.

The primary source of data was the patient files kept at the hypertension clinic and the pharmacy register to complement the information. The data abstraction tool was pretested on 10 patient files and standardized before being used for actual data collection and these 10 patient files were not part of the final patient files considered. Demographic, clinical and health system data was collected by two trained registered clinic nurses. The Principal Investigator (PI) regularly double-checked the filled data abstraction tool by trained registered record officers against the patient files to ensure that the data collected was accurate and free of errors.

For the qualitative component of the study, to enhance reflexivity phone interviews were conducted by two trained records officers who were familiar with the hypertension clinic environment but were not directly involved in clinical decision-making for the participants. This minimized undue influence while allowing rapport to be established. The Principal Investigator (PI), a clinical epidemiology scholar with training in qualitative research, supervised data collection and provided oversight to minimize interviewer bias. The two trained records officers contacted confirmed lost to follow up patient's and guided them through the informed consent process. The participants were recruited from 20th January 2024–28th January 2024. Those who provided consent were interviewed by phone to explore the reasons for their LTFU. The interviews were conducted using an interview guide developed in both English and Luganda, based on participant language preference (S2 Appendix). Interviews were audio recorded with prior permission from the participants and were held in the records office, in the presence of the Principal Investigator (PI). Each interview lasted a maximum of 30 minutes. To minimize loss of information, field notes were also taken in notebooks during each session. Interviews continued until thematic saturation was reached, defined as the point when no new codes or insights emerged, which occurred after 16 interviews. To ensure the credibility and dependability of the findings, patient verification and peer-review quality control practices were employed. Transcripts were not returned to participants for feedback due to nature of interviewed participants.

## Data management

The collected quantitative data was entered into EpiData Manager version 4.7.0, where it was verified for accuracy, consistency, and completeness. Variable-level missingness was minimal (<1%); missing values (DBP at last visit) were imputed using the respective measure of central tendency, and missing age values were computed from dates of birth of the patients. It was then cleaned and edited before being analyzed using Stata version 15.0. For qualitative data, audio recordings in English and Luganda were transcribed verbatim and translated into English text before analysis. Both qualitative and quantitative data were securely stored on a password-protected computer to ensure data confidentiality.

## Data analysis

Descriptive analysis where measures of central tendency and dispersion (mean, standard deviation, median and inter-quartile ranges (IQR)) were computed for numerical variables. For categorical variables, frequencies and percentages were computed, and tables and graphs were used to visualize the analyzed data.

A proportion was obtained to determine the percentage of patients that experienced attrition by dividing the number of hypertensive patients who were lost to follow-up by the study sample size.

The probabilities of patients staying in care at different intervals of the follow-up period were determined using the Kaplan-Meier method, and the log-rank test was used to determine the significance of observed differences between groups. The proportional hazards assumption was evaluated using both graphical methods and the Schoenfeld residuals test which was statistically significant (p = 0.0096) (S3 Appendix), indicating that the proportional hazards assumption was violated for the model overall. In particular, the test revealed that last-visit DBP also violated the assumption (p = 0.0015) hence the model extended to include time varying covariates. Both baseline and last-visit SBP and DBP measurements were included in the model. Baseline SBP and DBP and last-visit SBP, which met the proportional hazards assumption, were modeled as fixed covariates representing patients' initial and most recent clinical status.

Variables with p-values of ≤ 0.2 at bivariate analysis and those known to be associated with attrition from literature were considered for multivariate analysis. A chunk test was used to compare the reduced and full model and therefore assess for interaction (S4 Appendix). Confounding was then assessed where a variable was considered a confounder if the change in the hazard ratio (HR) was > 10%. Hazard ratios (HR), p-values and the 95% confidence intervals were reported.

Qualitative data analysis involved the utilization of a thematic analysis codebook, applying the 6 phases inherent in the thematic analysis (TA) approach [22].The data was transcribed, translated in English, coded, and synthesized using Open Code version 4.02 to yield notable themes [23]. Three trained data coders independently coded the transcripts to enhance reliability and reduce individual coder bias. A thematic analysis codebook was applied, and themes were inductively derived from the data. Discrepancies in coding were resolved through discussion and consensus among the coders. Analytical rigor was ensured through peer debriefing with the research team and systematic documentation of coding decisions. Data saturation was reached after the 16th interview when no new codes or insights emerged, and an illustrative table (S1 Table) presents each theme with representative participant quotations. Participants did not provide feedback on the findings.

## Ethical considerations

Permission to conduct the study was granted by the Clinical Epidemiology Unit (CEU). Ethical approval was obtained from the Institutional Review Board (IRB) through the School of Medicine Research and Ethics Committee (SOMREC) of Makerere University College of Health Sciences (Mak-SOMREC-2023–584). Additionally, for retrospective records review, SOMREC approved a waiver of consent to allow the use of patient records. Written informed consent was obtained from participants confirmed to be lost to follow-up, who took part in the qualitative phase of the study. Trained registered record officers, together with the principal investigator, reviewed patient files, ensuring that any information extracted was kept

strictly confidential and not disclosed to third parties. Anonymity was maintained by omitting identifying variables such as names during data extraction. Participants were provided with detailed information about the study in English or Luganda, and confidentiality was further ensured through anonymized transcripts and secure, password-protected data storage accessible only to the research team.

## Results

Among the 1,278 patients registered at clinic, 63 were excluded due to missing more than 30% of study data. A total of 1215 patient files meeting the eligibility criteria were selected for the quantitative study. Of these, 20 participants were selected to take part in the in-depth phone interviews (Fig 1).

### Characteristics of patients registered at the clinic from January 2020 to December 2022

More than half of the patients resided outside Kampala (58.8%, n = 714) and majority were female (76.1%, n = 924), with mean age of 56.5 years (SD 13.9). The majority did not smoke (99.4%, n = 1208), while 98.2% had no history of alcohol use (98.2%, n = 1193). About half had a comorbidity (50.8%, n = 617), while 92.8% (n = 1128) had never been hospitalized

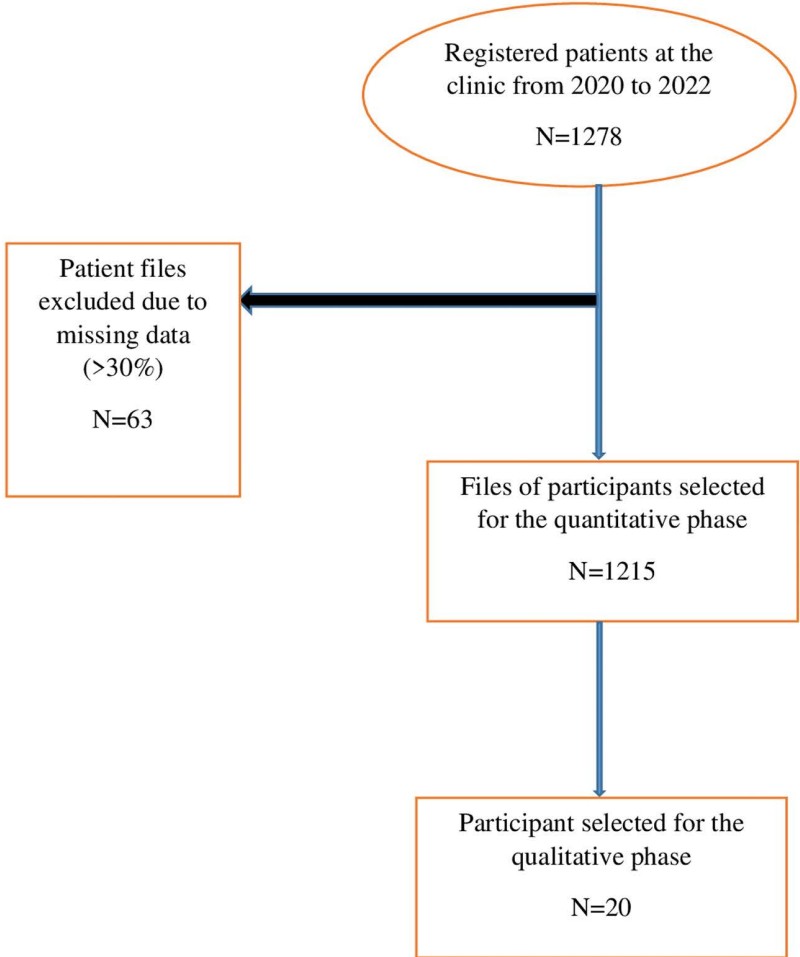

**Fig 1. Study profile.**

and 58.5%(n = 711) had experienced side effects. The median baseline SBP (1st, 3rd quartile) was 150 mmHg (138, 168) and the median baseline DBP was 87 mmHg (77, 97).

Regarding the drug regimen, the majority were on combination therapy (86.3%, n = 1048) and the least on diuretics (0.08%, n = 10). Most of the patients (59.6%, n = 724) had prescriptions of ≤ 2 antihypertensive drugs. Medicine stock outs were reported in 80.3% (n = 922) of the patients during at least one of the visits.

A small part of the patients used herbal medicine (1.0%, n = 12). On the last clinic visit, the median SBP was 145 (132, 159), and the median DBP was 83 (75, 92). Business was the most common occupation among the patients (29.1%, n = 188) followed by peasants (24.6%, n = 159) (Table 1).

### Attrition among patients registered at the clinic from Jan 2020 and Dec 2022

Of the 1,215 patients, 690 experienced attritions, resulting in an overall attrition proportion of 56.8% (95% CI: 54.0–59.6). The median duration of follow-up was approximately 16 months. Patients registered in 2020 had the highest attrition (64.9%), followed by 2022 (55.0%), and 2021 had a relatively reduced attrition (54.7%) When considering sex, male patients had a greater attrition (64.6%) than females (54.3%). Furthermore, patients residing outside Kampala had a relatively higher attrition (59.8%,) than those dwelling within Kampala, where the attrition was 52.6%. Kaplan-Meier survival analysis revealed statistically significant differences in survival probabilities based on sex (p < 0.0001), place of residence (p = 0.0001), and cohort entry year (p < 0.0001) (Fig 2).

### Factors associated with time to attrition among patients between January 2020 and December 2022

In bivariate analysis factors such as residence, cohort entry year, sex, smoking, age, side effects, baseline SBP, drug regimen, SBP on the last visit, and DBP on the last visit were selected for multivariate analysis (Tables 2 and 3). In multivariate analysis factors associated with attrition were age (HR = 0.904, 95% CI 0.877–0.932), residence: Outside Kampala (HR = 1.311,95%CI 1.121–1.533), Sex: Female (HR = 0.713, 95% CI 0.602–0.845), 2022 cohort entry year (HR = 1.433, 95% CI 1.156–1.777), SBP on last visit (HR = 1.013,95% CI 1.008–1.017) and DBP on last visit (HR = 0.957,95% CI 0.925–0.990). DBP on the last visit was a time-varying covariate with p = 0.006 (Table 4).

### Description of the participants that participated in the phone interviews

There were 20 patients out of the 690 found to be lost to follow up that were selected for the interviews, however only 16 interviewed. The mean age of the participants was 50 years (SD 11.99). 9 (56.3%) of the participants were female and 11 (68.8%) of the participants were residing outside Kampala. 5 (31.5%) of the participants had a comorbidity, 11 (68.8%) of the participants had experienced drug related side effects and 2 (12.5%) of participants reported to have been hospitalized in the past (Table 5).

### Underlying reasons for loss to follow up among patients lost to follow up

Participants reported a variety of factors contributing to LTFU from hypertension care. These were grouped into three major themes: [1] Structural and Contextual Barriers, [2] Health System Barriers, and [3] Illness Perceptions and Health-Related Limitations. Each major theme encompassed several subthemes (Fig 3), with each subtheme supported by illustrative participant quotations presented (S1 Table).

### Theme 1: Structural and contextual barriers

**Preference for alternative source of medication.** Participants obtained medication from nearby private clinics and pharmacies, with some preferring herbal remedies as alternatives to conventional medicine ''*I'm also a health work, most*

**Table 1. Characteristics of patients registered at the clinic from Jan 2020-Dec-2022 (n = 1215).**

| Variable | Category | Number (N) | Percentage (%) |
|---|---|---|---|
| Residence | Kampala | 501 | 41.2 |
| | Outside Kampala | 714 | 58.8 |
| Cohort entry year | 2020 | 231 | 19.0 |
| | 2021 | 502 | 41.3 |
| | 2022 | 482 | 39.7 |
| Sex | Male | 291 | 24.0 |
| | Female | 924 | 76.0 |
| Mean age (SD) | | 56.5 (13.9) | |
| Smoking status | Yes | 7 | 0.6 |
| | No | 1208 | 99.4 |
| Alcohol use status | Yes | 22 | 1.8 |
| | No | 1193 | 98.2 |
| Comorbidity | Yes | 617 | 50.8 |
| | No | 598 | 49.2 |
| Hospitalization | Yes | 87 | 7.2 |
| | No | 1128 | 92.8 |
| Side effects | Yes | 711 | 58.5 |
| | No | 504 | 41.5 |
| Baseline SBP | | | |
| Median (1st, 3rd quartile) | | 150 (138,168) | |
| Baseline DBP | | | |
| Median (1st, 3rd quartile) | | 87 (77,97) | |
| Drug regimen | Combination | 1048 | 86.3 |
| | CCB | 72 | 05.9 |
| | ACE-I or ARBs | 52 | 04.3 |
| | Beta blocker | 33 | 02.7 |
| | Diuretics | 10 | 00.8 |
| Number of antihypertensives | ≤2 | 724 | 59.6 |
| | >2 | 491 | 40.4 |
| Medicine stock-out | Yes | 922 | 80.3 |
| | No | 226 | 19.7 |
| Herbal medicine use | Yes | 12 | 01.0 |
| | No | 1203 | 99.0 |
| Last visit SBP | | | |
| Median (1st, 3rd quartile) | | 145 (132,159) | |
| Last visit DBP | | | |
| Median(1st, 3rd quartile) | | 83 (75,92) | |
| Marital status | Single | 7 | 00.6 |
| | Married | 448 | 36.9 |
| | Other | 760 | 62.5 |
| Occupation(n = 647) | Business | 188 | 29.0 |
| | Peasant | 159 | 24.6 |
| | House wife | 122 | 18.9 |
| | Health worker | 59 | 09.1 |
| | Teacher | 26 | 04.0 |
| | Engineer | 18 | 02.8 |

*(Continued)*

 

**Table 1.** (Continued)

| Variable | Category | Number (N) | Percentage (%) |
|---|---|---|---|
| | Driver | 13 | 02.0 |
| | Security | 11 | 01.7 |
| | Other | 51 | 07.9 |

CCB-Calcium Channel Blockers, ACE-I-Angiotensin-Converting Enzyme Inhibitor, ARBs- Angiotensin II Receptor Blockers and Other (Waitress, Evangelist, Designer, Journalist, shopkeeper, Office assistant, banker, Surveyor and carpenter).

*of the time I get my drugs from somewhere else, I buy from a pharmacy nearby home…….”* (**Female, LTFU**), *“…….my sibling at Najjanakumbi gave me some herbal medicine, that I use………”* (**Female, LTFU**).

**Financial hardship.** Economic hardship was a key deterrent to continued care as participants could not afford transportation to the clinic *“……. I've been having a challenge of lack of money; I'll harvest some maize to see if I can generate funds to come …….”* (**Male, LTFU**).

**Transportation barriers exacerbated by COVID-19 restrictions and geographical distance.** Participants reported that pandemic-related restrictions and long travel distances to the clinic affected their ability to attend appointments.

*“……the problem is COVID-19 came in and destabilized movement, even public means we use was stopped…….”* (**Female, LTFU**), *“……. I met some people and they told me to go to Kasese hospital for treatment because the distance to Mulago was far……...”* (**Male, LTFU**).

**Social disruptions and emotional strain.** Participants reported bereavement, prolonged travel for burial ceremonies, and lack of support or caretakers as key reasons for missing hypertension clinic follow-up visits. *“……I had some problems; I lost my relative and I traveled for burial and took long to come back…...”* (**Female, LTFU**), *“……I don't have a caretaker, it is me who takes care of myself, my children who would take care of me are not around…….”* (**Female, LTFU**).

**Limited mobility due to advanced age.** Elderly participants reported mobility limitations as a reason for their loss to follow up. *“……. being an elderly person and weak, I was tired and decided to just sit home and leave alone with going to the clinic……”* (**Female, LTFU**).

**Competing work demands and fixed clinic schedules.** Work-related obligations conflicted with non-flexible clinic schedules, led to LTFU

*“… work is too much at the specialized hospital where I work, a few times I visited clinic people at work complained thinking I had gone to work somewhere else…”* (**Female, LTFU**).

### Theme 2: Health system barriers

**Overcrowding and long waiting times.** Overcrowding and waiting for long before served at the clinic discouraged participants from returning for follow-up visits. *“……. sometimes you reach at the clinic and you are made to stay in the queue for so long…….”* (**Female, LTFU**), *“…...I came to the clinic, there were very many patients and we would spend a lot of time there…….”* (**Female, LTFU**).

**Perceived medical rudeness and unfriendly provider attitudes.** Negative interactions with healthcare providers led to dissatisfaction and disengagement.

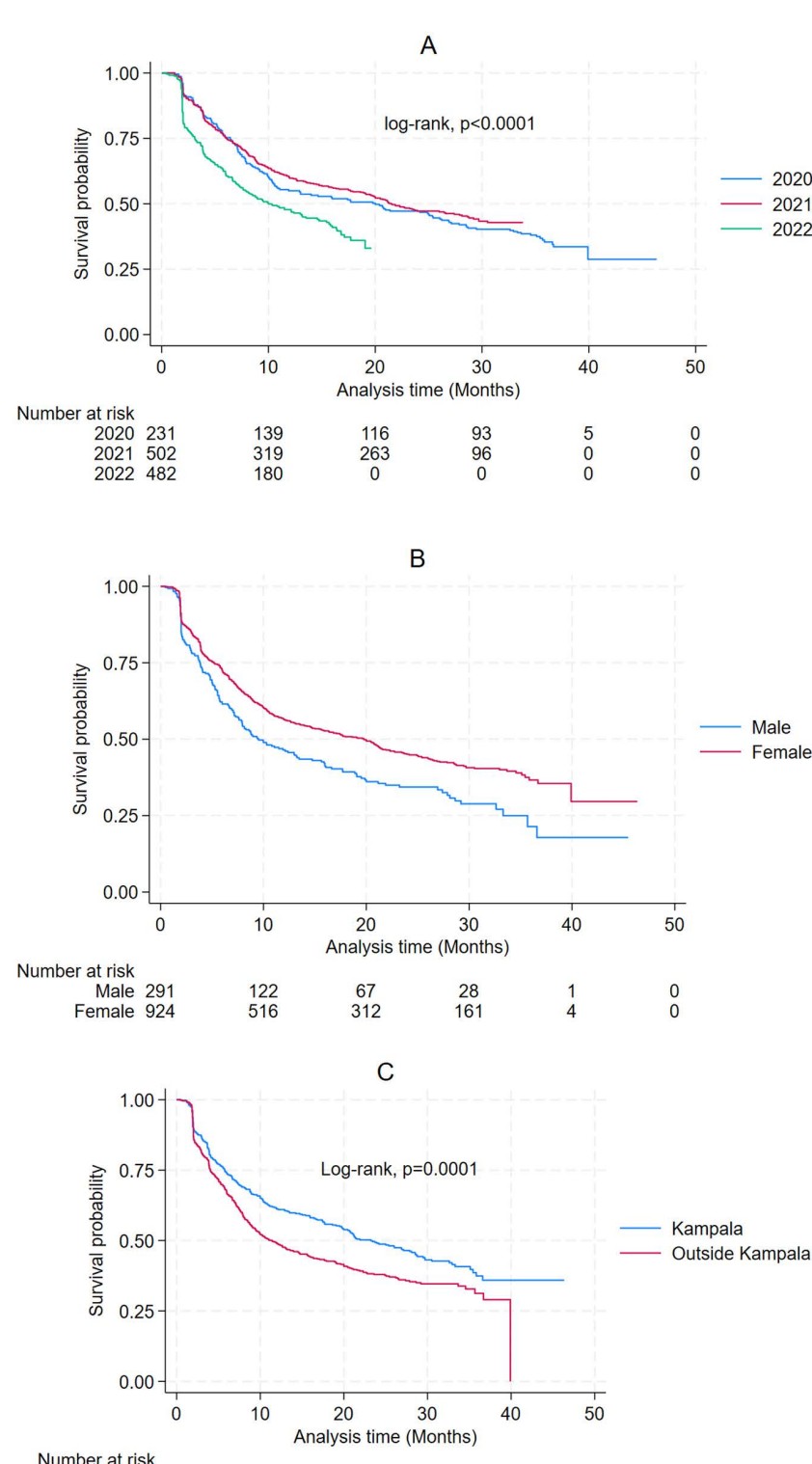

**Fig 2. Kaplan Meier survival curves for attrition among patients between Jan 2020 and Dec 2022 (A-Cohort entry year, B-Sex, C-Residence).**

**Table 2. Bivariate analysis for patient related factors associated with time to attrition among 1215 patients between Jan 2020-Dec 2022.**

| Variable | HR (95% CI) | p-value |
|---|---|---|
| **Residence** | | |
| Kampala | Ref | |
| Outside Kampala | 1.374 (1.176-1.606) | <0.001 |
| **Sex** | | |
| Male | Ref | |
| Female | 0.706 (0.597-0.835 | <0.001 |
| **Age (per 1 year)** | 0.999 (0.993-1.004) | 0.637 |
| **Smoking status** | | |
| Yes | Ref | |
| No | 0.682 (0.305-1.523) | 0.351 |
| **Alcohol use** | | |
| Yes | Ref | |
| No | 0.896 (0.528-1.521) | 0.685 |
| **Occupation** | | |
| Business | Ref | |
| Peasant | 0.650 (0.551-1.042) | 0.245 |
| House wife | 0.980 (0.678-3.050) | 0.201 |
| Health worker | 0.450 (0.310- 2.890) | 0.254 |
| Teacher | 0.720 (0.525-1.150) | 0.325 |
| Engineer | 0.691 (0.615-1.810) | 0.980 |
| Driver | 1.011 (0.434-1.431) | 0.421 |
| Security | 1.510 (0.781-3.100) | 0.267 |
| Other | 0.890 (0.343-2.194) | 0.723 |

HR= hazard ratio, CI=confidence interval.

*"…….. most of the doctors are rude, you ask them a question and he is rude and doctors are not willing to help…."* (**Female, LTFU**).

**Recurrent stockouts of prescribed medication.** Inconsistent availability of medication at the clinic pharmacy discouraged patients from returning. *"……even when you get transport money, you don't find medicine…."* (**Female, LTFU**)

**Frustration with appointment scheduling and inaccessible care.** Appointment frustration was a major reason participant missed follow-ups, as scheduling and securing timely visits were challenging. *"……I was told to go see some doctor but I couldn't find him for three weeks I came……"* (**Male, LTFU**).

### Theme 3: Illness perceptions and health-related limitations

**Physical limitations due to hypertension-related complications.** Physical impairments resulting from complications of hypertension also impeded attendance of clinic visits.

*"…...I stopped coming because I had a stroke and so it was hard for me to move…."* (**Male, LTFU).**

**Perceived lack of treatment effectiveness.** Some participants lost confidence in their treatment regimen, expressing doubts about its effectiveness. *"……. I would not get any change after getting the medication……"* (**Female, LTFU**)

Table 3. Bivariate analysis for clinical and health facility related factors associated with time to attrition among 1215 patients between Jan 2020-Dec 2022.

| tvc | | | | |
| --- | --- | --- | --- | --- |
| Variable | HR (95% CI) | p-value | HR (95% CI) | p-value |
| **Cohort entry year** | | | | |
| **2020** | **Ref** | | | |
| **2021** | **0.925 (0.754-1.135)** | **0.455** | | |
| **2022** | **1.500 (1.213-1.855)** | **< 0.001** | | |
| **Comorbidity** | | | | |
| Yes | Ref | | | |
| No | 1.149 (0.989-1.334) | 1.149 | | |
| **Hospitalization** | | | | |
| Yes | Ref | | | |
| No | 0.798 (0.533-1.193) | 0.271 | | |
| **Side effects** | | | | |
| Yes | Ref | | | |
| No | 1.134 (0.974-1.320) | 0.107 | | |
| **Baseline SBP(per 1** mmHg) | 1.002 (0.999-1.005) | 0.178 | | |
| **Baseline DBP(per 1** mmHg) | 1.000 (0.995-1.005) | 0.934 | | |
| **Drug regimen** | | | | |
| CCB | Ref | | | |
| Combination | 0.769 (0.568-1.042) | 0.090 | | |
| ACE-1 or ARBs | 0.692 (0.427-1.122) | 0.135 | | |
| Beta blockers | 1.381 (0.852-2.239) | 0.190 | | |
| Diuretics | 0.825 (0.352-1.934) | 0.658 | | |
| **Number of antihypertensives** | | | | |
| ≤2 | Ref | | | |
| >2 | 0.925 (0.794-1.077) | 0.315 | | |
| **Medicine stock out** | | | | |
| Yes | Ref | | | |
| No | 0.878 (0.714-1.080) | 0.217 | | |
| **Herbal use** | | | | |
| Yes | Ref | | | |
| No | 1.646 (0.683-3.969) | 0.267 | | |
| **Last visit SBP(per 1** mmHg) | 1.009 (1.006-1.013) | <0.001 | | |
| **Last visit DBP(per 1** mmHg) | 1.015 (1.006-1.023) | 0.001 | 0.998 (0.998 −0.999) | <0.001 |

HR= hazard ratio, CI=confidence interval, tvc= time varying covariates.

**Perceived wellness and the absence of symptoms.** Several participants discontinued follow-up because they felt physically well and did not perceive a need for ongoing care. *"…. if I'm feeling well, is there need to come back to the clinic…."* (**Female, LTFU**).

### Integration of quantitative and qualitative findings

Quantitative analysis identified younger age, male sex, residence outside Kampala, entry in the 2022 cohort, higher systolic blood pressure at the last visit, and lower diastolic blood pressure at the last visit as predictors of attrition. Qualitative findings provided context to these associations. Patients living outside Kampala reported long travel distances, high transport

**Table 4. Multivariate analysis for predictors of time to attrition among 1215 patients between Jan 2020-Dec 2022.**

| Variable | HR (95% CI) | p-value | HR (95% CI) | P-value |
|---|---|---|---|---|
| | | | tvc | |
| **Residence** | | | | |
| Kampala | Ref | | | |
| Outside Kampala | 1.244 (1.063- 1.455) | 0.006 | | |
| **Sex** | | | | |
| Male | Ref | | | |
| Female | 0.734 (0.620- 0.869) | <0.001 | | |
| **SBP on last visit (per 1 mmHg)** | 1.014 (1.009-1.018) | <0.001 | | |
| **DBP on last visit (per 1 mmHg)** | 0.957 (0.925-0.990) | 0.011 | 0.9989(0.9981-0.9997) | 0.006 |
| **Age (per 1 year)** | 0.947(0.931-0.963) | <0.001 | | |
| **Cohort entry year** | | | | |
| **2020** | Ref | | | |
| **2021** | 0.937 (0.763-1.151) | 0.537 | | |
| **2022** | 1.433 (1.156-1.777) | 0.001 | | |

HR = hazard ratio, CI = confidence interval, tvc = time varying covariates. Continuous covariates are expressed per 1-unit increase, age (per 1 year) and SBP/DBP, per 1 mmHg).

**Table 5. Description of the 16 participants that took part in the phone interview.**

| Characteristics | Category | n (%) |
|---|---|---|
| Sex | Male | 7 (43.8) |
| | Female | 9 (56.2) |
| Age | | |
| Mean (SD) | | 50 (12.0) |
| Residence | Kampala | 5 (31.3) |
| | Outside Kampala | 11 (68.7) |
| Presence of comorbidity | Yes | 5 (31.3) |
| | No | 11 (68.7) |
| Presence of side effects | Yes | 11 (68.7) |
| | No | 5 (31.3) |
| Hospitalization | Yes | 2 (12.5) |
| | No | 14 (87.5) |

n-number of participants, %- percentage.

costs, and competing livelihood demands, which contributed to disengagement. Male patients described prioritizing work obligations over clinic attendance, reflecting gendered health-seeking behaviors. Elevated systolic blood pressure was linked to attrition, and interviews revealed that perceptions of poor disease control or side effects led to frustration and withdrawal from care. In contrast, lower diastolic blood pressure was protective, consistent with accounts that feeling clinically stable encouraged continued follow-up. The higher attrition observed in the 2022 cohort coincided with health system challenges, including frequent medicine stock-outs and COVID-19 restrictions, which patients described as discouraging.

Overall, attrition was shaped not only by clinical status but also by structural, health system, and psychosocial factors. The joint display demonstrates how statistical predictors align with patient narratives, underscoring the need for retention strategies that address both measurable risks and the lived realities of patients (S2 Table).

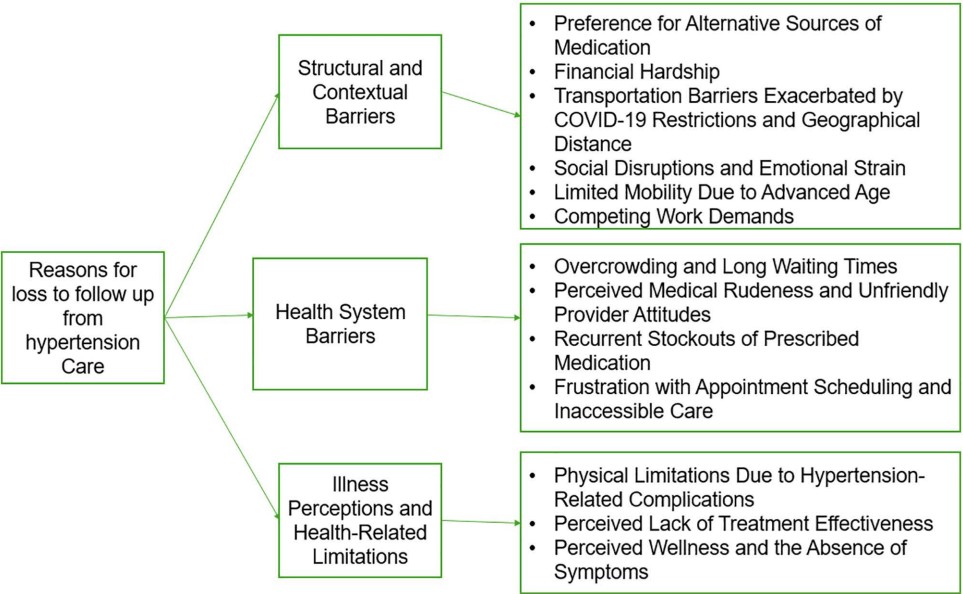

**Fig 3. Coding tree.**

## Discussion

The study found an attrition proportion of 56.8% indicating that slightly more than half of the patients on chronic antihypertensive were lost to follow-up during the study period. This suggests that the retention is quite low to achieve the WHO targeted treatment goal of bp < 140/90 mmHg in all patients with hypertension [24]. These results were consistent with those reported previous studies. Engelland and colleagues among 4403 hypertensive patients, reported attrition proportion of 51% [25]. Similarly, Ramsay and colleagues observed a 50% attrition proportion among 40 hypertension patients attending a hypertension clinic over 15 months [26]. MA and colleagues reported > 30% attrition in a study of 520 hypertensive patients in China [11]. These comparable findings highlight the widespread challenge of retaining hypertensive patients in long term care.

However, these results were different from the studies by Meena and colleagues conducted among a group of 1036 hypertensive patients who were registered at the NCD clinic in Pratap Nagar, Jodhpur, India and Kassavou and colleagues conducted among 101 hypertensive people in a primary health care which reported a lower attrition proportion of 9.2% and 7.92% respectively among people with hypertension [27,28]. The difference in the attrition proportions could be due to the disparity in the study designs, sample size and study setting.

Attrition among patients was highest in the cohort entry year 2020, which coincides with the onset of the COVID-19 pandemic in Uganda. The country reported its first confirmed case in March 2020, and soon after, strict public health measures were implemented, including nationwide lockdowns, travel restrictions, curfews, and limitations on public transportation [29]. These measures, while necessary to curb viral transmission, significantly hindered patients' ability to access routine healthcare services.

The patient's place of residence was statistically associated with time to attrition among hypertensive patients. Patients who resided outside Kampala had 24.4% higher risk over time of loss to follow up compared to those who resided in Kampala. This might be due to the fact that individuals residing in Kampala had easy access to the clinic, being in close proximity to it. This explanation aligns with the qualitative findings, where living far away was identified as one of the key reasons for patients being lost to follow-up while undergoing chronic antihypertensive treatment. Similarly, A. Baldé and

colleagues concurred in their findings, indicating that residing more than 5 km away from a healthcare facility was linked to an increased risk of LTFU among individuals living with HIV in Mali, a comparable chronic condition [30].

Patient sex was statistically associated with time to attrition among hypertensive patients. Female patients had a 26.6% lower risk over time of loss to follow up than their male counterpart. These findings are consistent with studies by Degoulet and colleagues conducted among 1346 medical records of hypertensive patients in Paris, and Hernandez and colleagues conducted among 6677 patients with hypertension or diabetes in Cambodia. [31,10] This could be because women are more likely to seek medical attention than the men [32]. However a study by Given and colleagues [33] found Sex not to be a significant predictor of attrition. This might stem from the fact that Given and colleagues [33] conducted a randomized controlled trial with a small sample size (153) compared to this study.

Age was significantly associated with time to attrition among patients with hypertension. For every 1-year increase in age of the patient, there was 5.3% decrease in time to attrition. Elderly patients tend to be concerned about their health compared to young individuals who exhibit less interest. This finding is in agreement with what was reported by Hernandez and colleagues in a study conducted in Cambodia among antihypertensive patients [13]. However, this finding diverges from our qualitative aspect which found advanced age as a reason for LTFU. The disparity could be due to the fact that most of the participants in the qualitative study were adults and most elderly.

Blood pressure measurements on the last visit date to the clinic were significantly associated with time to attrition. For every 1 mmHg increase in SBP at the last visit to the clinic, there was 1.4% increase in time to attrition. Additionally, for every 1 mmHg increase in DBP at the last visit to the clinic, there was 4.3% decrease in time to attrition and at any given time as it increases, the time to attrition decreases by 0.11%. Higher systolic blood pressure may prompt closer clinical monitoring or greater perceived risk, which could delay attrition, whereas elevated diastolic blood pressure may be perceived as less severe, leading to earlier disengagement. DBP levels change over time due to lifestyle modification, inconsistent adherence to medication and treatment adjustments by health care providers. While last-visit SBP and DBP were included in the model, residual bias may persist because last-visit measurements could have been recorded close to the attrition event

The calendar year a patient was registered at the clinic was significantly associated with time to attrition. Patients who were registered in the year 2022 had a 43.3% higher risk over time of attrition compared to those that were registered in 2020, possibly reflecting post-COVID health system adjustments. By 2022, although COVID-19 cases had subsided, changes in clinic operations and staffing, or follow-up procedures may have created new barriers to retention, increasing the likelihood of LTFU among this cohort.

Drug regimen was not associated with time to attrition in this study. While no previous studies have directly examined this relationship among hypertensive patients, evidence from chronic care settings suggest that medication regimens can influence retention. For example drug regimens have been associated with attrition among 58,115 people living with HIV (PLHIV) in China by Zhu and colleagues [34]. Although both hypertension and HIV require long term medication adherence, the populations differ in terms of treatment complexity, perceived severity, stigma and intensity of follow up. These contextual differences may explain why drug regimen influenced attrition in the HIV study but not in our hypertensive population.

Medicine stock out was not associated with time to attrition. However, a study conducted by Pasquet and colleagues in Abidjan, Côte d'Ivoire among HIV infected patients, a similar chronic illness found drug stock to be associated with interruption from care [35]. The findings from the qualitative aspect are in agreement with these findings as medicine stock-out was highlighted by participants as one of the reasons for LTFU.Several other factors including number of antihypertensive drugs, history of hospitalization, drug related side effects and smoking status were not associated with time to attrition. The absence of hypertension specific comparative studies limits direct contextualization of these findings. While some studies in other chronic disease populations have reported associations between similar factors, differences in disease burden, care models, medication complexity, and follow up schedules restrict their comparability [13,10,36,37].These

 

findings therefore contribute new evidence indicating that these commonly hypothesized factors may play a limited role in attrition within hypertensive care setting.

The qualitative findings reveal a multidimensional interplay of why patients get lost to follow-up, categorized into structural and contextual barriers, health system barriers, and illness perceptions and health-related limitations. Structural and contextual issues such as financial hardship, transportation challenges especially heightened by COVID-19 restrictions and competing work demands illustrate the broader socioeconomic constraints that hinder continuity of care, consistent with findings from prior studies in developed and low-resource settings [38–40]. Health system barriers, including long waiting times, perceived provider rudeness, and frequent medication stockouts, reflect systemic inefficiencies that erode patient trust and motivation, these findings are similar to what is reported in South Africa and in a narrative review conducted by systematically searching electronic databases [41,42]. Additionally, patients' perceptions of being well and symptom-free, coupled with physical limitations, diminish their perceived necessity for ongoing care, consistent with findings from studies in the UK, USA, and Spain on low risk perception in asymptomatic hypertension [43–45].

## Study strength and limitation

The mixed-methods design provided comprehensive insights by integrating quantitative and qualitative findings. The qualitative component complemented and clarified the quantitative results, enriching the understanding of patient attrition and its associated factors.

However, this study had several limitations. Missing data from retrospective patient files, including variables such as education level, may have introduced unmeasured confounding. Consecutive sampling in the quantitative phase could have led to selection bias, limiting generalizability. Recall bias was possible in the qualitative phase, as participants might have forgotten or misreported past experiences. Moreover, the observational design restricts causal inference; thus, the identified relationships should be interpreted as associations rather than causal effects.

## Conclusions

This study found a substantial attrition of 56.8%, with the highest number of patients lost to follow-up occurring in 2020, coinciding with the peak of the COVID-19 pandemic. Various factors were associated with time to attrition, including age, male sex, residing outside the capital city, and last visit BP measurements and cohort entry year. Furthermore, LTFU was driven by structural and contextual barriers, health system challenges, and illness perceptions and health-related limitations. These included financial hardship, long distances, COVID 19 restrictions, overcrowding, provider attitudes, and perceived lack of treatment benefit.

These findings imply that retention rates are below the Centers for Disease Control and Prevention's 80% retention target [46], highlighting the need to implement targeted strategies to enhance patient retention, particularly among high risk groups. Strengthening tracking systems, improving access to healthcare, and minimizing the effects of external disruptions such as pandemics may support sustained patient engagement.

## Supporting information

**S1 Appendix. Detailed methodological sample size formulas for survival analysis.**
(PDF)

**S2 Appendix. Interview guide.**
(PDF)

**S3 Appendix. Graphical and statistical approaches used to assess for Cox PH assumptions.**
(PDF)

**S4 Appendix. Assessing for interaction.**
(PDF)

**S1 Table. Illustrative quotes linking each theme to representative participant responses.**
(PDF)

**S2 Table. Joint Display Linking Quantitative Predictors to Qualitative Themes.**
(PDF)

## Acknowledgments

We extend our sincere appreciation to the hypertension clinic staff for their invaluable support throughout this study, and to all the participants for their willingness to take part.

## Author contributions

**Conceptualization:** Nathan Ntenkaire, Julius Mubiru.

**Data curation:** Nathan Ntenkaire, Maxwell Okwero.

**Formal analysis:** Nathan Ntenkaire, Patience Muwanguzi, Maxwell Okwero, Beatrice Basuuta.

**Funding acquisition:** Nathan Ntenkaire.

**Methodology:** Nathan Ntenkaire, Brian Mikka, Sandra Lunkuse, Maxwell Okwero, Beatrice Basuuta.

**Project administration:** Nathan Ntenkaire.

**Resources:** Nathan Ntenkaire.

**Supervision:** Nathan Ntenkaire, Mark Kaddu Mukasa, Patience Muwanguzi, Joan N. Kalyango.

**Writing – original draft:** Nathan Ntenkaire, Mark Kaddu Mukasa, Joan N. Kalyango.

**Writing – review & editing:** Nathan Ntenkaire, Mark Kaddu Mukasa, Patience Muwanguzi, Brian Mikka, Sandra Lunkuse, Julius Mubiru, Maxwell Okwero, Beatrice Basuuta, Douglas Bulafu, Joan N. Kalyango.

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
