## [Decision Letter · Decision Letter 0]

9 Oct 2025

Dear Dr. Ntenkaire,

Thank you for submitting your manuscript to PLOS ONE. After careful consideration, we feel that it has merit but does not fully meet PLOS ONE’s publication criteria as it currently stands. Therefore, we invite you to submit a revised version of the manuscript that addresses the points raised during the review process.

We look forward to receiving your revised manuscript.

Kind regards,

Ignatius Ivan, M.D

Academic Editor

PLOS ONE

Journal Requirements:

2. We note that there is identifying data in the Supporting Information file < Attrition_wdtaset.xlsx     >. Due to the inclusion of these potentially identifying data, we have removed this file from your file inventory. Prior to sharing human research participant data, authors should consult with an ethics committee to ensure data are shared in accordance with participant consent and all applicable local laws.

-Location data

Please remove or anonymize all personal information (<ID, Tel_NOK, Tel_patient and Age>), ensure that the data shared are in accordance with participant consent, and re-upload a fully anonymized data set. Please note that spreadsheet columns with personal information must be removed and not hidden as all hidden columns will appear in the published file.

**Additional Editor Comments:**

1. Methodological Clarity and Sampling Strategy

Issue: The sampling procedure and sample size calculation contain redundant explanations and inconsistent formulas (pages 5–7). The justification for using both Kish (1965) and survival data formulas is not clearly connected to study objectives.

Revision Suggestion: Clarify how the final sample size (n=1215) was derived and why it exceeds the calculated requirement. Streamline the sample size section by presenting a single, coherent formula with assumptions, parameters, and effect size justification.

2. Mixed-Methods Integration and Design Justification

Issue: Although the study uses a sequential explanatory mixed-methods design, the integration of quantitative and qualitative findings is not explicitly described in the Results or Discussion sections.

Revision Suggestion: Add a section explicitly describing how qualitative results complemented quantitative findings (e.g., a joint display or integrated interpretation paragraph). State how qualitative insights influenced the interpretation of statistical results.

3. Statistical Rigor and Model Selection

Issue: The rationale for using an Extended Cox Regression Model is mentioned (violation of proportional hazard assumption) but not shown graphically or statistically. Furthermore, the proportional hazards test results are not reported.

Revision Suggestion: Include diagnostic plots or a supplementary figure/table demonstrating the proportional hazard assumption violation. Provide the name of the variable that violated the assumption and justify the model extension quantitatively.

4. Validity and Reliability of Qualitative Analysis

Issue: The qualitative phase describes thematic analysis with independent coders, but there is no mention of intercoder reliability or saturation confirmation process beyond “data saturation was achieved after 16 interviews.”

Revision Suggestion: Specify whether intercoder agreement was measured (e.g., Cohen’s kappa) and describe the analytical validation process (peer debriefing, triangulation, or audit trail). Add an illustrative quote table linking each theme to representative participant responses.

5. Overinterpretation of Quantitative Findings

Issue: The conclusion implies that factors such as age and male sex cause attrition, which overstates the observational design. Phrases like “patients who resided outside Kampala were 31% more likely to get lost to follow-up” may imply causality rather than association.

Revision Suggestion: Revise language to emphasize associations and avoid causal implications. For example, use “was associated with” instead of “led to” or “caused.” Include a clear statement acknowledging the limitations of observational inference.

6. Contextualization with Existing Literature

Issue: The discussion references multiple unrelated chronic disease studies (e.g., HIV, diabetes) without sufficient justification for their analogy to hypertension care. Regional literature from sub-Saharan Africa on hypertension retention is underrepresented.

Revision Suggestion: Strengthen discussion by citing and comparing results with regionally relevant hypertension adherence studies (e.g., Nigeria, Kenya, Tanzania). Discuss context-specific barriers such as drug stockouts or urban–rural disparities using LMIC evidence.

7. Writing Structure and Presentation

Issue: The manuscript is lengthy and contains redundant phrases (e.g., repeated mention of “Mulago hospital” and “chronic antihypertensive therapy”). Tables are numerous but lack clarity on the meaning of abbreviations, and figure legends (e.g., Fig 2) are insufficiently described.

Revision Suggestion: Streamline narrative flow by merging repetitive sentences and ensuring all abbreviations are defined upon first use. Improve figure/table captions for stand-alone comprehension. Move extensive methodological formulas and supplementary data into appendices.

Reviewers' comments:

Reviewer's Responses to Questions

**Comments to the Author**

1. Is the manuscript technically sound, and do the data support the conclusions?

Reviewer #1: Yes

Reviewer #2: Partly

Reviewer #3: Yes

2. Has the statistical analysis been performed appropriately and rigorously?

Reviewer #1: Yes

Reviewer #2: No

Reviewer #3: I Don't Know

3. Have the authors made all data underlying the findings in their manuscript fully available?

Reviewer #1: Yes

Reviewer #2: Yes

Reviewer #3: No

4. Is the manuscript presented in an intelligible fashion and written in standard English?

Reviewer #1: Yes

Reviewer #2: Yes

Reviewer #3: Yes

Reviewer #1: This is an automated report for PONE-D-25-33575. This report was solicited by the PLOS One editorial team and provided by ScreenIT.

ScreenIT is an independent group of scientists developing automated tools that analyze academic papers. A set of automated tools screened your submitted manuscript and provided the report below. Each tool was created by your academic colleagues with the goal of helping authors. The tools look for factors that are important for transparency, rigor and reproducibility, and we hope that the report might help you to improve reporting in your manuscript. Within the report you will find links to more information about the items that the tools check. These links include helpful papers, websites, or videos that explain why the item is important. While our screening tools aim to improve and maintain quality standards they may, on occasion, miss nuances specific to your study type or flag something incorrectly. Each tool has limitations that are described on the ScreenIT website. The tools screen the main file for the paper; they are not able to screen supplements stored in separate files. Please note that the Academic Editor had access to these comments while making a decision on your manuscript. The Academic Editor may ask that issues flagged in this report be addressed. If you would like to provide feedback on the ScreenIT tool, please email the team at ScreenIt@bih-charite.de. If you have questions or concerns about the review process, please contact the PLOS One office at plosone@plos.org.

Reviewer #2: Summary

This manuscript reports a sequential explanatory mixed-methods study of attrition among patients on chronic antihypertensive therapy at Mulago Hospital, Uganda. The quantitative component is a retrospective cohort using routinely collected clinic records; the qualitative component comprises interviews with patients to contextualize disengagement from care. The topic is important and relevant for NCD program performance in a low-resource setting. The dataset appears sufficiently large and the mixed-methods design is appropriate. With several clarifications and analytic refinements, the work can meet PLOS ONE’s standard of technical soundness with transparent reporting.

Major comments (please address in revision)

1. Outcome definition & time origin. Attrition is defined as ≥2 consecutive missed appointments. Please specify the usual appointment interval (e.g., monthly/bi-monthly) and the exact time origin for survival analyses (e.g., clinic registration or first antihypertensive prescription) so readers can understand person-time construction and the practical meaning of “two consecutive” misses.

2. Survival modelling details. You report Kaplan–Meier and Cox (with proportional hazards checks and use of an extended Cox model when needed). Please: (a) state which variables violated PH and how you addressed this; (b) provide PH diagnostics (tests/plots) in the supplement; (c) describe model specification clearly candidate variables, criteria for entry/retention, handling of interactions and confounding.

3. Temporal alignment of blood pressure covariates. SBP/DBP measured at “last visit” are used as predictors. If serial measures exist, consider modelling SBP and DBP as time-varying covariates to avoid bias from values recorded close to the attrition event. If only fixed values are used, justify this choice and discuss limitations; a sensitivity analysis using baseline or penultimate-visit values would be helpful.

4. Scaling and interpretation of continuous predictors. Report the units for continuous covariates in the model (e.g., age per 1 or 10 years; SBP/DBP per 1 or 10 mmHg) and ensure the text, tables, and conclusions reflect the chosen scaling.

5. “Rate” vs “proportion.” Where you present the overall percentage disengaged, please refer to it as an attrition proportion/percentage. Reserve “rate” for person-time metrics (events per person-months/years) if you also estimate those.

6. Missing data. Beyond excluding records with high overall missingness, please describe variable-level missingness and your approach (complete-case, missing-indicator, or multiple imputation). If feasible, add a sensitivity analysis (e.g., multiple imputation) for key predictors.

7. Presentation of survival results. Add numbers-at-risk beneath Kaplan–Meier curves (or report in figure legends/tables), include log-rank p-values where curves are compared, and report median follow-up and total events to contextualize information content.

8. Qualitative reporting & integration. Methods are concisely described; to align with COREQ expectations, please expand on reflexivity (interviewer roles/position), sampling and recruitment, language/translation, and how thematic saturation was judged. A joint display linking quantitative predictors (e.g., distance/residence) to qualitative themes (transport costs, stock-outs, clinic processes) would strengthen integration and the discussion of mechanisms.

9. Calendar time and external shocks. Because disengagement may vary by year (e.g., service disruptions), consider modelling calendar period (or include it as a covariate) and/or discuss how temporal shocks could influence retention.

10. Contextual claims and citations. Where you compare observed retention/attrition to programmatic “targets,” please provide a supporting citation and clarify whether such targets apply specifically to hypertension care.

Minor comments (clarity, style, formatting)

• Ensure consistent terminology (e.g., “loss to follow-up” with hyphens; consistent use of BP/SBP/DBP).

• Standardize units and spacing (e.g., “mmHg”, “%”).

• In the setting description, use “intensive care” rather than “intensive” alone.

• Define all abbreviations at first use in the abstract and main text.

• Tables: label units and scaling in headers (e.g., “HR per 10 years of age,” “HR per 10 mmHg SBP”).

• Consider adding a simple flow diagram (records screened → eligible → included → retained vs LTFU/other outcomes).

Data, code, and reproducibility

The anonymized dataset provided in the Supporting Information is a strong step toward compliance with the PLOS Data policy. For maximal reproducibility, please also share the analysis code (e.g., do-files/scripts) and a brief data dictionary, ideally in a public repository with a stable link/DOI referenced in the Data Availability Statement.

Ethics

Ethical approvals and consent/waiver procedures are described. Please ensure the ethics section clearly distinguishes approvals for the retrospective records review and for the qualitative interviews (including consent procedures, language, and confidentiality safeguards).

Overall assessment

The research question is important, the design is appropriate, and the findings are potentially useful for strengthening hypertension services in similar settings. Addressing the analytic clarifications (survival modelling assumptions, covariate scaling and temporal alignment, missing-data handling), tightening terminology around “rate” vs “proportion,” and enhancing qualitative reporting/integration will substantially improve technical rigor and transparency. With these revisions, the manuscript should meet PLOS ONE’s criteria that conclusions be well supported by rigorously conducted and clearly reported analyses.

Reviewer #3: Thank you for this generally well written and interesting paper.

I have a couple of minor comments, and do believe the discussion needs to be looked into.

- Please provide the definition you used of Lost to follow up.

- Please provide an overview of variables extracted from patient files

- Please provide the topic guide and explain what topics were discussed during the interviews and if there was any theory the questions were based on.

- Please ensure LTFU is abbreviated the first time it's used and the abbreviation is used consistently.

- Please provide information about recruitment of participants for interviews: how were they identified, when and where were they asked to participate, what was sampling based on?

In terms of the discussion, the literature comparisons are not the clearest. There are some very long sentences with multiple studies addressed (see 344-350). Additionally, some of the explanations about why data you collected differed from literature data are insufficient, for example:

- Line 370: do you have a reference for this? Unsure what this is based on.

- Line 377 to 380: I am unsure what is meant here. Most patients being adult and elderly does not seem to be an explanation for age as a reason for loss to follow up. Please clarify.

- Line 381 to 387: This is unclear. Do you mean that time to attrition increases when systolic blood pressure increases, but decreases when diastolic blood pressure increases? That seems an interesting finding, why could that be? This is not discussed here, and the reasons provided don't seem very general, not specific for blood pressure measurements.

Line 388 to 392: You first mention PLWHIV is a similar population to your population, before explaining the difference in outcomes between your study and the study in China being due to the latter having a different population. Please rectify and provide additional reasons for the difference.

- Various discrepancies in the discussion section are explained due to the referenced studies having different populations. These studies may thus not be the best to compare your findings with and would be good to see if there are better suited studies out there, or maybe the data is not comparable as such and the paragraphs should be rewritten to reflect this.

Additionally the discussion is very long, particularly the comparisons with other scientific litertature is taking up a lot of space.

It would be good to reflect on some of your study strengths.

Conclusion

- the impact of COVID-19 is barely discussed in the discussion but mentioned in conclusion. Would be good to reflect on the impact of the pandemic in more detail in the discussion.

**Do you want your identity to be public for this peer review?** For information about this choice, including consent withdrawal, please see our Privacy Policy

Reviewer #1: No

Reviewer #2: No

Reviewer #3: No

---

## [Author Response · Author response to Decision Letter 1]

24 Nov 2025

Thank you for the thoughtful review and detailed feedback. We appreciate the comment and have addressed it in our revised submission. Although we updated the data availability section in the manuscript to include the public repository link, the portal version had not been updated. This has now been corrected and aligned. Thank you again for the opportunity to improve our work.

---

## [Decision Letter · Decision Letter 1]

26 Jan 2026

Attrition and associated factors among patients on chronic antihypertensive therapy at Mulago hospital, Uganda: A mixed method study

PONE-D-25-33575R1

Dear Dr. Nathan Ntenkaire

We’re pleased to inform you that your manuscript has been judged scientifically suitable for publication and will be formally accepted for publication once it meets all outstanding technical requirements.

Kind regards,

Ignatius Ivan, M.D

Academic Editor

PLOS One

Additional Editor Comments (optional):

Reviewers' comments:

Reviewer's Responses to Questions

**Comments to the Author**

Reviewer #2: All comments have been addressed

2. Is the manuscript technically sound, and do the data support the conclusions?

Reviewer #2: Yes

3. Has the statistical analysis been performed appropriately and rigorously?

Reviewer #2: Yes

4. Have the authors made all data underlying the findings in their manuscript fully available?

Reviewer #2: Yes

5. Is the manuscript presented in an intelligible fashion and written in standard English?

Reviewer #2: Yes

Reviewer #2: Thank you for your careful and comprehensive revisions to the manuscript. The improvements made in response to the previous round of reviews have substantially strengthened the methodological clarity, analytical rigor, and interpretive coherence of the study

**Do you want your identity to be public for this peer review?** For information about this choice, including consent withdrawal, please see our Privacy Policy

Reviewer #2: No

---

## [Editor Report · Acceptance letter]

PONE-D-25-33575R1

PLOS One

Dear Dr. Ntenkaire,

I'm pleased to inform you that your manuscript has been deemed suitable for publication in PLOS One. Congratulations! Your manuscript is now being handed over to our production team.

Kind regards,

on behalf of

dr. Ignatius Ivan

Academic Editor

PLOS One